# Safety, efficacy and health impact of electronic nicotine delivery systems (ENDS): an umbrella review protocol

Sonu Goel ,[1,2] Muhammed Shabil,[3] Jagdish Kaur ,[4] Anil Chauhan,[5] Arvind Vashishta Rinkoo[6]

¹University of Limerick, Limerick, Ireland
²Department of Community Medicine and School of Public Health, Post Graduate Institute of Medical Education and Research, Chandigarh, Chandigarh, India
³Department of Pharmacy Practice, M. S. Ramaiah University of Applied Sciences, Banglore, India
⁴World Health Organization Regional Office for South-East Asia, New Delhi, Delhi, India
⁵Department of Telemedicine, Post Graduate Institute of Medical Education and Research, Chandigarh, India
⁶World Health Organization Regional Office for South-East Asia, New Delhi, India

**Correspondence to**
Dr Sonu Goel; sonu.goel@ul.ie

## ABSTRACT

**Background** Electronic nicotine delivery systems (ENDS), commonly known as e-cigarettes or vapes, have witnessed a rise in popularity, particularly among the youth. Although they were initially introduced as an alternative to traditional smoking, the design and function of ENDS vary. The potential health effects of ENDS, especially in comparison to traditional cigarettes, are a matter of ongoing debate. Given the increasing number of clinical studies and systematic reviews on this topic, there exists a demand for an umbrella review that offers a comprehensive assessment. The goal of this study is to perform an umbrella review of systematic reviews and meta-analyses to assess the safety, efficacy, health implications and potential gateway effect associated with ENDS.

**Methods and analysis** This umbrella review will adhere to the Joanna Briggs Institute (JBI) framework and the Preferred Reporting Items for Systematic Review and Meta-Analysis guidelines. A planned literature search will be executed across databases such as OVID, PubMed/MEDLINE, EMBASE, Cochrane Library and Web of Science. The inclusion criteria are systematic reviews that discuss ENDS and e-liquids in the context of safety, efficacy and health outcomes. The exclusion criteria include narrative reviews, non-systematic reviews and studies not in English. Quality of the selected studies will be evaluated using the AMSTAR V.2 Scale. An overlap assessment will be done using the Corrected Covered Area, and data synthesis will be presented both narratively and in tabulated forms

**Ethics and dissemination** Ethics approval is not required for this study, as it does not involve the collection of original data. The results will be disseminated through peer-reviewed publication. The findings will offer crucial insights for stakeholders, policy-makers and the general public, underlining the health implications and the role of ENDS in tobacco cessation.

## STRENGTHS AND LIMITATIONS OF THIS STUDY

⇒ The umbrella review approach allows a thorough synthesis of existing reviews on electronic nicotine delivery systems, offering insights into safety, efficacy and health implications.
⇒ Adherence to the Joanna Briggs Institute framework and Preferred Reporting Items for Systematic Review and Meta-Analysis guidelines ensures methodological rigour and transparent review reporting.
⇒ The insights provided have practical relevance and applicability for stakeholders, policy-makers and the general public.
⇒ Excluding non-English studies may introduce language bias, overlooking significant findings in other languages.
⇒ Reliance on existing reviews means inherent gaps, limitations or biases in them will affect this study's conclusions.

## INTRODUCTION

Electronic cigarettes, commonly known as e-cigarettes or vapes, are devices designed to aerosolise a substance called 'e-liquid' for inhalation.[1–3] These devices, also known as electronic nicotine delivery systems (ENDS), were first created in 2003 with the intention of serving as a tool to help individuals quit smoking traditional cigarettes.[4] ENDS function by using a heating element to heat the e-liquid, producing a vapour that can be inhaled through a mouthpiece. During this process, new chemical compounds may be generated, some of which could pose health risks. E-cigarette devices come in various forms, ranging from older, lower-power models resembling traditional cigarettes (often called 'cigalikes') to refillable pens, larger tank systems and more recent innovations such as compact devices using high-concentration nicotine salt pods and disposable options.[3 5] E-cigarettes have gained widespread popularity and are used by millions of people globally, with a notable prevalence among younger individuals.[6 7]

The significantly reduced levels of harmful substances in ENDS compared with cigarettes have prompted researchers to explore their potential for assisting with smoking cessation.[8 9] However, concerns about the negative health impacts of secondhand aerosol exposure remain. The limited regulation of these products might also play a role in the

expansion of the ENDS market, where tobacco companies have a notable presence. This could potentially lead to a resurgence of smoking habits, undermining years of antitobacco efforts in the Southeast Asian Region. There has been a surge in clinical studies on ENDS, and the Cochrane Group published the first systematic review on ENDS in 2014.[10] In recent years, there has been an increase in the publication of systematic reviews and meta-analyses (SRMAs) covering various aspects of ENDS to assess their effectiveness in aiding tobacco cessation.[11–19]

The present circumstances necessitate mobilising policy-makers to address this issue in a region where a substantial burden of tobacco use is exacerbated by a significant population of susceptible young individuals and a limited well-established tobacco cessation resource. Because of the need for a comprehensive approach, an additional step in synthesising existing SRMAs has been established in the form of umbrella reviews. Umbrella reviews are conducted consistently, enabling a comprehensive analysis by integrating existing SRMAs. They swiftly assess abundant evidence, comparing prior systematic reviews and achieving coherence by subdividing complex issues into specific populations or interventions.[20] The purpose of this umbrella review is to evaluate the impact of ENDS on health and its efficacy and safety in tobacco cessation.

### Objectives

The aim of the present study is to conduct three umbrella reviews of SRMAs to evaluate safety, efficacy, health outcomes and gateway effect of ENDS. Three umbrella reviews will be conducted.

### Umbrella review 1

Objective 1: to assess the effectiveness of ENDS as a tool for tobacco cessation by investigating quit rates among tobacco smokers using ENDS.

Objective 2: to identify and analyse adverse effects associated with the use of ENDS.

### Umbrella review 2

Objective: to evaluate the short-term and long-term health outcomes linked to the utilisation of ENDS.

### Umbrella review 3

Objective: to explore the potential gateway effect of ENDS, particularly in relation to the initiation of tobacco smoking among individuals who were either never tobacco users or had previously quit smoking.

### METHODS
### Patient and public involvement

Patients or the public were not involved in the design, or conduct, or reporting, or dissemination plans of our research

The method for conducting this umbrella review is based on the framework set forth by the Joanna Briggs Institute (JBI).[20] We will consistently follow the Preferred Reporting Items for Systematic Review and Meta-Analysis

(PRISMA) guidelines throughout the entirety of our process. A checklist, derived from the PRISMA Protocols, has been filled out and is available in online supplemental table 1.[21] Our umbrella review protocol will be registered in the PROSPERO. Any adjustments to our methodology will be documented and thoroughly explained in the final umbrella review report. The study was started on 1 October 2023, and it will continue until July 2024.

### Inclusion and exclusion criteria

We will include systematic reviews in our evaluation, even if they form part of broader assessments. These broader assessments may cover topics such as the safety and efficacy of ENDS for tobacco smoking cessation or reduction, health-related outcomes associated with ENDS use in humans and the potential initiation of tobacco smoking by never smokers or former tobacco users due to ENDS (referred to as the 'gateway effect').

The interventions include e-cigarettes, ENDS and e-liquids. Notably, non-nicotine e-cigarettes and other pharmacological treatments, such as nicotine replacement therapy, are excluded.

For assessing efficacy, quitting rates of combustible tobacco smoking among those who are using ENDS will be considered. These rates can range from a period of 1 month to 1 year, based on data from primary studies. Reduction in the number of cigarettes smoked, reduction in the number of cigarette-smoking days and proportion of participants achieving a 50% reduction in tobacco smoking during the follow-up period will be considered. The risk of continuing tobacco smoking in both the intervention and comparison groups will also be evaluated.

For safety considerations, we will assess any adverse events linked to e-cigarettes. This includes but not limited to incidents such as poisoning, explosions and health issues due to malfunctioning ENDS, as well as allergic reactions to any contained chemicals. Health outcomes are categorised into short-term and long term. Short-term refers to immediate outcomes, and long term encompasses outcomes observed over a span of months to years. The health outcomes of interest are:

1. Incidence and risk of clinical disease endpoints such as coronary artery disease, myocardial infarction, congestive heart failure, stroke, other cardiovascular diseases and any type of cancer.
2. Development of risk factors and intermediate biological effects on health, including atherosclerosis, high blood pressure, lung damage, elevated glucose levels and dyslipidaemia.
3. Incidence and risk of respiratory diseases, oral health complications, renal health concerns, neurological effects, optical health issues, impaired wound healing, olfactory issues, endocrine problems, allergic reactions and haematological outcomes.
4. Pregnancy-related risks, neonatal effects, developmental and reproductive issues and corresponding changes in clinical parameters.

5. Mental health effects, and the impact on sleep patterns, quality and duration.

All the health outcomes will be evaluated based on proportion, risk or mean difference of clinical parameters.

To evaluate the potential 'gateway effect' of ENDS, the incidence and risk of initiation of combustible tobacco cigarette smoking in non-smokers or former smokers who use ENDS will be considered.

Our inclusion criteria will extend to systematic reviews that incorporate observational studies or randomised controlled trials (RCTs). Additionally, we will consider more recent primary studies that have not been previously incorporated into existing systematic reviews. These primary studies will be RCTs, cohort, case–control, non-randomised clinical trials, cross–sectional studies. We will exclude narrative reviews, non-systematic reviews, commentaries and editorial articles. Additionally, studies not available in English language or published in non-peer-reviewed journals will be excluded. The specific criteria for the population, intervention, comparator and outcome (PICO) are detailed in table 1, providing a comprehensive framework that delineates the scope of our umbrella review.

### Databases and searching

We will conduct a comprehensive systematic literature search across the various databases such as OVID, PubMed/MEDLINE, EMBASE, Cochrane Library, CINAHL and Web of Science. The search strategy will be optimised to enhance accuracy and comprehensiveness, if necessary. The search will be conducted by an experienced medical librarian. A search strategy for PubMed is given in table 2. Keywords and MeSH terms related to ENDS will be used in the search process. The search will employ a study design filter to identify systematic reviews, whenever available, within each database. A language filter for the English will be applied. We will not impose any date limits on the search, ensuring that we capture relevant literature spanning various time frames. Reviewers will examine the citations of the included articles to identify additional relevant articles.

### Screening and selection

The search results will be consolidated and deduplicated using Covidence. Screening processes will be conducted using Covidence for the initial screening of titles and abstracts, followed by full-text screening. Two independent reviewers will be responsible for evaluating the titles, abstracts and full texts of articles to determine their eligibility. Systematic reviews that align with the predefined PICO criteria will be included in the analysis. In cases of uncertainty or disagreement between the two reviewers, a third reviewer will assess the article to reach a consensus and make a final determination regarding its inclusion or exclusion.

### Data extraction

Data extraction will be carried out by two different reviewers independently, and a prepiloted and standardised data extraction form by JBI will be used. In cases where there are disagreements in data extraction, a third reviewer will be consulted to facilitate discussion and reach a consensus. Both quantitative and qualitative from each included study will be extracted. The extracted information will be displayed in a tabular format for clear and concise presentation accompanied by explanatory text. The quantitative compilation of results will include details such as the first author's name, publication year, study setting, the number of RCTs and observational studies encompassed in the systematic review, characteristics of the study participants, specifics of the interventions and comparators employed, and the outcomes assessed. This will also cover the total number of participants, effect size with their CIs, metrics and results for heterogeneity, results pertaining to publication bias and the tests used and the type of quality assessment tool implemented along with its results. Additionally, values for the total pooled effects, Cochran Q statistic, Egger's test and $I^2$ will be extracted.

Additionally, information regarding the funding sources of funding for systematic reviews and any potential conflicts of interest, especially concerning financial benefits related to the intervention, will also be extracted. A data extraction form template is given as online supplemental table 2.

### Quality assessment

The quality assessment of included SRMAs will be conducted by two reviewers independently using the AMSTAR V.2 Scale.[22] A third reviewer will be consulted in case of difference of opinion. AMSTAR V.2 consists of 16 domains, 7 of which are classified as critical domains because of their substantial impact on confidence in the conclusions drawn from systematic reviews.[22] These critical domains encompass a range of crucial aspects, including the registration of the review protocol, appropriateness of the search strategy, reason for excluding specific studies, risk-of-bias assessment of the included studies and its influence on systematic review's conclusions, method used for evidence synthesis and consideration of publication bias. The systematic review's overall confidence level in its results will be categorised into four distinct levels: high, moderate, low and very low.[22]

### Data synthesis

Prior to conducting the synthesis of findings, a sensitivity analysis will be carried out to assess the extent of overlap among primary studies across the systematic reviews. This analysis is essential as overlapping studies can lead to potential biases in the analysis. For this purpose, we have chosen the Corrected Covered Area (CCA) metric,[23] a validated and widely used approach. To compute CCA, a citation matrix of primary studies will be created and this matrix will be included in our review. The CCA Score will

**Table 1** PICO

| PICO | Inclusion criteria | Exclusion criteria |
|---|---|---|
| Population | General population with or without cigarette smokers with >12 years of age | Animals<br>In vitro<br>In vivo |
| Intervention | E-cigarettes, electronic nicotine delivery systems, e-liquids | Nicotine replacement therapy<br>Non-nicotine e-cigarettes<br>Other pharmacological interventions |
| Comparison | For safety and efficacy:<br>Placebo e-cigarette (without nicotine) or any comparator treatment or combination of treatments usually given for smoking cessation, for example, nicotine replacement therapy, cigarette smokers without any treatment, without e-cigarette<br>For health outcomes:<br>Never smokers (no e-cigarette or combustible tobacco products ever)<br>Smoker populations—if no other comparator available for some outcomes<br>For gateway effect:<br>Never smoke, never e-cigarette users | Dual users of e-cigarette and tobacco |
| Outcome | Primary outcomes:<br>1. Tobacco smoking cessation, 50% reduction in cigarette consumption, adverse events<br>2. Clinical disease endpoints, such as myocardial infarction, coronary artery disease, congestive heart failure, stroke, other cardiovascular disease and cancer<br>3. Development of risk factors and intermediate biological effect of health outcomes such as atherosclerosis, high blood pressure, lung damage, high glucose levels, dyslipidaemia<br>4. Respiratory diseases oral health, renal health neurological effects optical health, wound healing, olfactory, endocrine, allergic diseases and haematological outcomes<br>5. Effect on pregnancy, neonatal effects, development and reproductive effects<br>6. Mental health, effects on sleep pattern, quality, duration<br>7. Gateway effect (ever smoking combustible tobacco cigarettes)<br>8. Nicotine dependency<br>9. Serious and non-serious adverse effects | Economic outcomes<br>Environmental outcomes |
| Study type | Systematic reviews and meta-analyses of RCTs and observational studies<br>Primary studies (observational studies and RCTs) | Case reports, non-human studies |
| Setting | Any country | No exclusion |
| Follow-up | No restrictions | No exclusion |
| Language | English | Not available in English |

PICO, population, intervention, comparator and outcome; RCTs, randomised controlled trials.

**Table 2** Search strategy for PubMed

| Search | Query | Results |
|---|---|---|
| #4 | Search: ((((("Electronic Nicotine Delivery Systems"[Mesh]) OR ("Vaping"[Mesh])) OR (("e-cig*" OR "ecig*" OR "e cig*" OR "electronic cig*" OR "electronic nicotine*" OR vape OR vapes OR vaporizer OR vapourizer OR vaporiser OR vapouriser OR vaper OR vapers OR vaping OR e-liquid OR ENDS))) OR ((E Cigarettes) OR (E-Cigarette) OR (E Cigarette) OR (Electronic Cigarette) OR (Cigarette, Electronic) OR (Cigarettes, Electronic))) AND ((((((("Adolescent"[Mesh]) OR ("Adult"[Mesh] OR "Young Adult"[Mesh])) OR (young people)) OR (middle aged)) OR (older adult)) OR (older people))) AND ((((systematic review) OR (systematic reviews)) OR (meta analysis)) OR (network meta analysis)) | 1564 |
| #3 | Search: (((systematic review) OR (systematic reviews)) OR (meta analysis)) OR (network meta analysis) Sort by: Most Recent | 468 238 |
| #2 | Search: ((((("Adolescent"[Mesh]) OR ("Adult"[Mesh] OR "Young Adult"[Mesh])) OR (young people)) OR (middle aged)) OR (older adult)) OR (older people) | 9 046 337 |
| #1 | Search: ((("Electronic Nicotine Delivery Systems"[Mesh]) OR ("Vaping"[Mesh])) OR (("e-cig*" OR "ecig*" OR "e cig*" OR "electronic cig*" OR "electronic nicotine*" OR vape OR vapes OR vaporizer OR vapourizer OR vaporiser OR vapouriser OR vaper OR vapers OR vaping OR e-liquid OR ENDS))) OR ((E Cigarettes) OR (E-Cigarette) OR (E Cigarette) OR (Electronic Cigarette) OR (Cigarette, Electronic) OR (Cigarettes, Electronic)) | 367 626 |

be categorised into different levels, including very high (>15), high,[11–15] moderate[6–10] and slight (0–5).[24] This analysis will help us manage and account for potential overlap among the primary studies.

The synthesis of evidence will be presented in both narrative and tabular formats. We will provide a table detailing the specifics of each systematic review included in our analysis, encompassing information such as the intervention studied, the target population, outcomes assessed, comparator, the number of primary studies and participants involved, the search databases used along with their respective dates and the effect estimates reported, such as risk ratios (RR), ORs, HR, mean difference (MD), Standardised MD or similar metrics when available and their CIs, heterogeneity, publication bias, final findings, quality assessments and a summary of the risk of bias identified in the primary studies included. A narrative approach will be used to summarise the evidence for each outcome, and we will also employ tabular formats where applicable to enhance clarity. Weightage will be given to the results of highest rated systematic review by AMSTAR V.2 where the reviews show a higher level of overlap. In instances where there are discrepancies between the results of recent and high-quality systematic reviews, we will conduct a reanalysis of the primary data to arrive at a conclusion.

If newer primary studies necessitate data synthesis, a meta-analysis will be performed if more than three studies are available for each outcome. Meta-analysis will be conducted by pooling the effect size from each study using a random effects model. The R software will be employed for the meta-analysis. Subgroup analyses will be undertaken based on variables such as participants, outcomes or any other relevant factors that might contribute to heterogeneity. The $I^2$ statistic will be used to assess heterogeneity. A p value of less than 0.05 will be deemed statistically significant. Publication bias will be assessed through visual inspection of funnel plot symmetry and the Egger's test. Sensitivity analysis will be performed to determine the effect of each study on the overall result.

The quality of evidence will be determined using Grading of Recommendations, Assessment, Development, and Evaluations criteria. Garding will be performed by considering five domains including risk of bias in the individual studies, inconsistency, indirectness, imprecision and publication bias for each outcome.[25] We will grade the strength of evidence (very low, low, moderate and high).

### Ethics and dissemination

Ethics approval is not required for this study as it does not involve the collection of original data. The results will be disseminated through peer-reviewed publication. The findings will offer crucial insights for stakeholders, policymakers and the general public, underlining the health implications and the role of ENDS in tobacco cessation.

**Contributors** SG is the guarantor. SG, JK and AVR conceptualised the topic. AC and MS analysed and finalised the methods. MS and AC drafted the manuscript. SG, JK, AVR and AC reviewed, edited and proofread the final draft. SG, MS, JK, AVR, and AC finalised and approved the manuscript.

**Funding** The review is funded by WHO with registration no: 2023/1386892-0, purchase order: 203214074.

**Disclaimer** The opinions or views expressed in this article are solely those of the authors and do not express the views or opinions of the organization to which the authors are affiliated.

**Competing interests**  None declared.

**Patient and public involvement**  Patients and/or the public were not involved in the design, or conduct, or reporting, or dissemination plans of this research.

**Patient consent for publication**  Not applicable.

**Provenance and peer review**  Not commissioned; externally peer reviewed.

**ORCID iDs**
Sonu Goel http://orcid.org/0000-0001-5231-7083
Jagdish Kaur http://orcid.org/0000-0001-9800-9203

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
