## [Reviewer comments · BMJ Open]

ARTICLE DETAILS

TITLE (PROVISIONAL)	Safety, efficacy and Health impact of Electronic Nicotine Delivery Systems (ENDS): An Umbrella Review protocol
AUTHORS	Goel, Sonu; Shabil, Muhammed; Kaur, Jagdish; Chauhan, Anil; Rinkoo, Arvind Vashishta

VERSION 1 – REVIEW

REVIEWER	Mehrotra, Ravi National Institute of Cancer Prevention and Research
REVIEW RETURNED	11-Nov-2023

GENERAL COMMENTS	The authors of the protocol for the umbrella review detail the proposed methodology to assess the safety, efficacy, health implications, and potential gateway effect associated with Electronic Nicotine Delivery Systems using the standard PRISMA and JBI strategies, which will be registered in PROSPERO. The proposed databases include OVID, PubMed/MEDLINE, EMBASE, Cochrane Library, and Web of Science. The protocol's search terms and possible limitations, including the exclusion of non-English publications, are mentioned. Inclusion and exclusion criteria are reasonable and the analysis will follow the PICO format. The quality assessment will be assured by utilising AMSTAR-2 criteria and quality of evidence using GRADE criteria. An overlap assessment will be done using the and data synthesised to come to objective and evidence-backed conclusions. The reviewer agrees with the proposed strategies.
--

REVIEWER	Freeman, Becky University of Sydney, School of Public Health
REVIEW RETURNED	14-Nov-2023

GENERAL COMMENTS	My biggest concern with this proposed protocol is that it includes four wide-ranging objectives – which given the breadth and scope of the literature seems too much for one umbrella review. I think the objectives could be grouped in to three separate umbrella reviews: Umbrella review 1 Objective 1: To assess the effectiveness of ENDS as a tool for tobacco cessation by investigating quit rates among tobacco smokers using ENDS. There are also a high number if systematic reviews of e-cigs as a smoking cessation tool which all draw on the very same limited
--

	number of clinical trails – how will this proposed protocol reflect this issue? Umbrella review 2 Objective 2: To identify and analyze adverse effects associated with the use of ENDS. Objective 3: To evaluate the short-term and long-term health outcomes linked to the utilization of ENDS. This umbrella review has recently been published - which also incorporates aspects of Objective 4: https://www.mja.com.au/journal/2023/218/6/electronic-cigarettes-and-health-outcomes-umbrella-and-systematic-review-global Umbrella review 3 Objective 4: To explore the potential gateway effect of ENDS, particularly in relation to the initiation of tobacco smoking among individuals who were either never tobacco users or had previously quit smoking. This related review has been previously published by BMJ Open: https://pubmed.ncbi.nlm.nih.gov/33785493/ I think it's will also be important to reflect how the proposed review will incorporate/complement the WHO ENDS reviews on these same topics.
--	---

VERSION 1 – AUTHOR RESPONSE

Reviewer: 1

Prof. Ravi Mehrotra, National Institute of Cancer Prevention and Research

Comments to the Author:

The authors of the protocol for the umbrella review detail the proposed methodology to assess the safety, efficacy, health implications, and potential gateway effect associated with Electronic Nicotine Delivery Systems using the standard PRISMA and JBI strategies, which will be registered in PROSPERO. The proposed databases include OVID, PubMed/MEDLINE, EMBASE, Cochrane Library, and Web of Science. The protocol's search terms and possible limitations, including the exclusion of non-English publications, are mentioned. Inclusion and exclusion criteria are reasonable and the analysis will follow the PICO format. The quality assessment will be assured by utilising AMSTAR-2 criteria and quality of evidence using GRADE criteria. An overlap assessment will be done using the and data synthesised to come to objective and evidence-backed conclusions. The reviewer agrees with the proposed strategies.

Response: Thank you for reviewing the protocol.

Reviewer: 2

Dr. Becky Freeman, University of Sydney

Comments to the Author:

My biggest concern with this proposed protocol is that it includes four wide-ranging objectives – which given the breadth and scope of the literature seems too much for one umbrella review.

I think the objectives could be grouped in to three separate umbrella reviews:

Umbrella review 1

Objective 1: To assess the effectiveness of ENDS as a tool for tobacco cessation by investigating quit rates among tobacco smokers using ENDS.

There are also a high number of systematic reviews of e-cigs as a smoking cessation tool which all draw on the very same limited number of clinical trials – how will this proposed protocol reflect this issue?

Umbrella review 2

Objective 2: To identify and analyze adverse effects associated with the use of ENDS.

Objective 3: To evaluate the short-term and long-term health outcomes linked to the utilization of ENDS.

This umbrella review has recently been published - which also incorporates aspects of Objective 4: <https://www.mja.com.au/journal/2023/218/6/electronic-cigarettes-and-health-outcomes-umbrella-and-systematic-review-global>

Umbrella review 3

Objective 4: To explore the potential gateway effect of ENDS, particularly in relation to the initiation of tobacco smoking among individuals who were either never tobacco users or had previously quit smoking.

This related review has been previously published by BMJ Open:

<https://pubmed.ncbi.nlm.nih.gov/33785493/>

I think it's will also be important to reflect how the proposed review will incorporate/complement the WHO ENDS reviews on these same topics.

Response: Thank you for reviewing the protocol. We acknowledge the comments from the learned reviewer.

We have discussed the broad scope of the review topic and the potential for conducting three umbrella reviews, as you suggested, with members of the WHO. As you mentioned, we have decided to proceed with these three umbrella reviews. Thank you for your suggestion.

We are aware that there are existing umbrella reviews covering aspects of e-cigarettes. However, our preliminary investigation has identified several additional systematic reviews, approximately 50, most of which were published in the past 2-3 years. These systematic reviews were not included in the previous umbrella reviews, as their searches mostly spanned up to 2020/2021. The number of reviews incorporated in these earlier studies is significantly less than we identified in our preliminary surface search. Conducting an updated umbrella review will provide beneficial and updated evidence for new policy-making and decision-making, as commissioned by the WHO.